# Minimally Invasive Off-Pump Coronary Artery Bypass as Palliative Revascularization in High-Risk Patients

**DOI:** 10.3390/clinpract15080147

**Published:** 2025-08-06

**Authors:** Magdalena Rufa, Adrian Ursulescu, Samir Ahad, Ragi Nagib, Marc Albert, Rafael Ayala, Nora Göbel, Tunjay Shavahatli, Mihnea Ghinescu, Ulrich Franke, Bartosz Rylski

**Affiliations:** 1Department of Cardiovascular Surgery, Robert Bosch Hospital, 70376 Stuttgart, Germany; adrian.ursulescu@rbk.de (A.U.); samir.ahad@rbk.de (S.A.); ragi.nagib@rbk.de (R.N.); marc.albert@rbk.de (M.A.); rafael.ayala@rbk.de (R.A.); nora.goebel@rbk.de (N.G.); tunjay.shavahatli@rbk.de (T.S.); mihnea.ghinescu@rbk.de (M.G.); bartosz.rylski@rbk.de (B.R.); 2Department of Cardiovascular Surgery, University Heart Centre Freiburg Bad Krozingen, 79189 Freiburg, Germany; ulrich.franke@uniklinik-freiburg.de

**Keywords:** minimally invasive direct coronary artery bypass (MIDCAB), minimally invasive multivessel coronary artery bypass grafting (MICS CABG), coronary artery bypass grafting (CABG), coronary artery disease (CAD), multivessel coronary artery disease (MV CAD), palliative therapy

## Abstract

**Background:** In high-risk and frail patients with multivessel coronary artery disease (MV CAD), guidelines indicated complete revascularization with or without the use of cardiopulmonary bypass (CPB) bears a high morbidity and mortality risk. In cases where catheter interventions were deemed unsuitable and conventional coronary artery bypass grafting (CABG) posed an unacceptable perioperative risk, patients were scheduled for minimally invasive direct coronary artery bypass (MIDCAB) grafting or minimally invasive multivessel coronary artery bypass grafting (MICS-CABG). We called this approach “palliative revascularization.” This study assesses the safety and impact of palliative revascularization on clinical outcomes and overall survival. **Methods:** A consecutive series of 57 patients undergoing MIDCAB or MICS-CABG as a palliative surgery between 2008 and 2018 was included. The decision for palliative surgery was met in heart team after carefully assessing each case. The patients underwent single or double-vessel revascularization using the left internal thoracic artery and rarely radial artery/saphenous vein segments, both endoscopically harvested. Inpatient data could be completed for all 57 patients. The mean follow-up interval was 4.2 ± 3.7 years, with a follow-up rate of 91.2%. **Results:** Mean patient age was 79.7 ± 7.4 years. Overall, 46 patients (80.7%) were male, 26 (45.6%) had a history of atrial fibrillation and 25 (43.9%) of chronic kidney disease. In total, 13 patients exhibited a moderate EuroSCORE II, while 27 were classified as high risk, with a EuroSCORE II exceeding 5%. Additionally, 40 patients (70.2%) presented with three-vessel disease, 17 (29.8%) suffered an acute myocardial infarction within three weeks prior to surgery and 50.9% presented an impaired ejection fraction. There were 48 MIDCAB and nine MICS CABG with no conversions either to sternotomy or to CPB. Eight cases were planned as hybrid procedures and only 15 patients (26.3%) were completely revascularized. During the first 30 days, four patients (7%) died. A myocardial infarction occurred in only one case, no patient necessitated immediate reoperation. The one-, three- and five-year survival rates were 83%, 67% and 61%, respectively. **Conclusions:** MIDCAB and MICS CABG can be successfully conducted as less invasive palliative surgery in high-risk multimorbid patients with MV CAD. The early and mid-term results were better than predicted. A higher rate of hybrid procedures could improve long-term outcome in selected cases.

## 1. Introduction

The anticipated mortality rate for coronary artery bypass grafting (CABG) in the elderly population varies between 2.5% and 4.5% [1]. However, the results may vary considerably depending on the patient’s preoperative condition. Individuals with coronary conditions demonstrate a higher prevalence of comorbidities, diminished physiological reserves and greater frailty as a consequence of an ageing demographic [2].

The probability of an adverse result after heart surgery is increased in frail patients [3]. Currently, multiple frailty scales have been created to encompass related but unique categories that illustrate different elements of frailty [4]. An analysis of eight commonly employed frailty tools demonstrated a significant absence of consensus among them [4]. Consequently, optimal long-term outcomes may only be attained by interdisciplinary collaboration aimed at identifying at-risk patients, improving their preoperative conditions, minimising surgical incisions and expediting postoperative recovery [3].

Nevertheless, in the majority of research, individuals with significant risk factors are infrequently represented, resulting in a limited number of studies addressing the outcomes of high-risk patients. The growing incidence of elderly or frail individuals suffering from severe coronary disease (CAD), who are deemed unsuitable for interventional coronary revascularisation, underscores the need to investigate alternative treatment modalities [5]. A more refined approach, such as minimally invasive direct coronary artery bypass (MIDCAB) grafting or minimally invasive multivessel coronary artery bypass grafting (MICS-CABG) via mini-thoracotomy, which circumvents both extracorporeal circulation and sternotomy may be better suited for this patient demographic. However, the great majority of these cases would receive an incomplete or as we named it ‘palliative’ revascularization.

Recent findings demonstrate that incomplete myocardial revascularization using off-pump left internal thoracic artery (LITA) to left anterior descending (LAD) grafting results in better outcomes than optimal medical therapy alone, especially in high-risk populations, concerning overall survival and cardiac-related mortality rates [6]. The LIMA-to-LAD graft may protect the heart from harmful left ventricular remodelling and reduce proarrhythmogenic foci by providing perfusion to the greatest extent of the myocardium [6,7]. As a result, revascularization of the LAD becomes a critical prognostic factor in patients with CAD [8,9].

This study assesses the safety and impact of MIDCAB and MICS-CABG as palliative revascularization on clinical outcomes and overall survival in a consecutive cohort of high-risk patients at a single centre.

## 2. Materials and Methods

A retrospective analysis was performed on high-risk patients with multivessel coronary artery disease for whom catheter interventions were inappropriate and conventional coronary artery bypass grafting presented an unacceptable perioperative risk due to advanced age, multimorbidity and frailty. These patients underwent either MIDCAB or MICS CABG surgery to minimise surgical burden and avoid extracorporeal circulation. Most patients were incompletely revascularized. We termed this method as “palliative revascularization,” with a primary emphasis on the revascularization of the anterior wall. Ultimately, we identified 57 cases treated between September 2008 and December 2018.

The primary objective of this research was to assesses the safety and impact of MIDCAB and MICS-CABG as palliative revascularization on clinical outcomes. The secondary aim included evaluating the incidence of major adverse cerebral and cardiovascular events (MACCE), as well as survival rate during the follow-up interval. The definition of MACCE includes acute myocardial infarction, stroke, recurrent revascularisation and mortality. Demographic and clinical characteristics, together with the in-hospital outcomes, were rigorously gathered and analysed. The employed surgical technique has been previously reported in detail [10].

Follow-up data were collected by mail or telephone interviews with the research participants or their referring cardiologists or general practitioners, should the person be unreachable. Information was also collected about the date and cause of death, when relevant.

### Statistical Analysis

Categorical variables are reported as absolute and relative frequencies, while continuous variables are expressed as mean ± standard deviation. To assess the differences between subgroups we used McNemar’s test for categorical variables. Differences were considered statistically significant if the *p*-value was less than 0.05.

The Kaplan–Meier test was employed to estimate patient survival rates. The statistical analysis was performed utilising IBM SPSS Statistics for Windows, Version 28.0, Armonk, NY: IBM Corp.

## 3. Results

### 3.1. Preoperative Characteristics

The mean age of the patients was 79.7 ± 7.4 years, and 80.7% of them were male. The youngest patient was 61 years old, while the oldest was 93 years old. In total, 13 patients exhibited a moderate EuroSCORE II, while 27 were classified as high risk, with a EuroSCORE II exceeding 5%. The mean EuroSCORE II value was 7.3 ± 6.8.

Atrial fibrillation was present in a significant number of individuals, specifically 45.6%. More than one-third of the patients had medically treated type II diabetes, and 28.1% had a positive history of peripheral vascular disease. The prevalence of chronic kidney disease was extremely high, at 43.9%. Three weeks anterior to the surgical procedure, 29.8% of the patients experienced an acute myocardial infarction. A total of 70.2% of patients had three-vessel disease at presentation, and 17.5% of procedures were deemed urgent (Table 1).

### 3.2. Intraoperative Course

A total of 48 procedures were performed as MIDCAB, while nine patients underwent MICS CABG involving two bypasses. All patients underwent LITA-LAD grafting. No conversions to sternotomy or cardiopulmonary bypass (CPB) occurred. The rate of completeness of revascularisation was notably low at 26.3%, with eight procedures categorised as hybrid, specifically reversed hybrid, involving percutaneous transluminal coronary angioplasty (PTCA) performed in an acute setting, followed by surgical revascularisation of the anterior wall weeks later (Table 2). Two patients underwent concurrent segmental lung resection, which histopathologic examination confirmed as malignancy.

Table 3 presents the significant data from the in-hospital postoperative course. Six patients, representing 10.5%, developed renal failure and required dialysis. Two patients experienced a cerebrovascular accident, representing 3.5% of the cohort. No reoperations were required for either bleeding or bypass revision.

There were four postoperative fatalities, one from cardiogenic shock and low cardiac output, one from a severe stroke and two from pneumonia; the 30-day mortality rate was 7%. In line with the advance directive signed before surgery and reinforced by their authorised family members, the therapy including vasopressors and all intensive care unit measures was stopped in all four patients who had an early cardiac death.

The average follow-up duration was 4.2 ± 3.7 years. The study included 57 cases, with four fatalities occurring within the initial 30 days. A follow-up rate of 91.2% was achieved after 48 patients were monitored; five cases were unfortunately not able to be reached and were thus deemed lost to follow-up. Survival rates at one-, three- and five-years were 83%, 67% and 61%, respectively. During the follow-up period, 33 major adverse cardiac and cerebrovascular events were recorded. Six myocardial infarctions, eight repeat revascularizations via PTCA, one stroke, and 24 fatalities constituted the causes of MACCE. Only two fatalities were confirmed to have a cardiac cause.

We chose to analyse whether the notably low revascularisation completion rate of 26.3% could have influenced the early and long-term outcomes (Table 4). None of the specified criteria achieved statistical significance, potentially attributable to the limited number of cases examined or the overall elevated risk of the cohort patients.

## 4. Discussion

The first step in improving the outcomes of patients who are frail and have multiple medical conditions is to gain an understanding of their condition at the time that surgical intervention is indicated. This recognition exercise would then allow physicians to advocate for the customization of the optimal perioperative pathway for each patient, particularly in the context of minimally invasive surgical options and postoperative care modifications [3,11].

In their extensive study involving 85 879 frail patients receiving isolated CABG, Dobaria et al. report a reduction in overall CABG volume and a rising prevalence of frailty within this cohort [11]. In light of this worry, Ponzi et al. advise limited aortic manipulation, minimal surgical incision and consideration of LIMA-LAD and PCI of other vessels as surgical principles in frail patients [3]. These ideas are implemented through procedures such as anaortic off-pump CABG, MIDCAB, MICS-CABG, robotic-assisted or fully robotic totally endoscopic coronary artery bypass (TECAB) and hybrid revascularisation [3]. Our work group fully concurs with their evaluation; nevertheless, there is no objective evidence or frailty assessment to substantiate the therapeutic decision in our cohort. Instead, it was based on individual judgement and interdisciplinary dialogue among all experts present in the heart team.

In our defence, it is important to point out that a large number of clinical instruments that are developed to evaluate frailty need active patient engagement. This is something that is not always possible for individuals who have compromised clinical, social and sometimes even educational backgrounds. In addition, these evaluations are frequently not feasible in cases that are considered to be urgent [3].

Given that our perspectives coincide with the aforementioned recommendations, we have resolved to implement a multidimensional frailty assessment in the future and to advocate more vigorously for completeness of revascularisation by hybrid methods.

This study reveals that for the selected patients, MIDCAB and MICS CABG demonstrated acceptable operative mortality and morbidity rates, which were lower than those predicted by EuroSCORE II, despite the high prevalence of significant coexisting risk factors.

Fraund-Cremer et al. conducted an analysis of 1 363 MIDCAB cases, with 53.1% presenting multivessel coronary artery disease (MV CAD). Their study indicated that patients with MV CAD had a significantly higher risk of postoperative stroke compared to those with less complex CAD [12]. This finding is consistent with our results, as both patients who suffered an in-hospital stroke and the one who reported a stroke during follow-up exhibited MV CAD. Furthermore, their research indicated, as anticipated, inferior long-term survival rates in MV CAD patients treated with MIDCAB compared to those with single-disease conditions, as well as in patients scheduled for hybrid procedures who did not receive the PTCA step [12]. The patients in their study exhibited a significantly lower EuroSCORE II of 1.7, in contrast to our study group, which had a mean EuroSCORE II of 7.3.

A study involving 35 high-risk patients treated with MIDCAB, of which 20 experienced incomplete revascularisation, indicated that MIDCAB was linked to low operative mortality and morbidity [2]. The one-year survival rate was 94%, and the follow-up assessment of LIMA graft patency and flow, conducted via exercise transthoracic Doppler echocardiography, yielded favourable results [2]. In comparison to our study group, their patients had a younger mean age of 69.2 years, and the reported EuroSCORE was 6.4 [2]. However, this is likely the initial version of EuroSCORE, as the study encompassed patients treated between 1998 and 2000.

When compared to the non-frail cohort, Dobaria et al. discovered that the frail cohort had approximately three times the odds of experiencing respiratory, cerebrovascular and renal problems during index hospitalisation [11]. Tran et al. conducted a comprehensive Canadian retrospective cohort study, revealing a significant association between frailty and long-term mortality, as well as a correlation between incomplete revascularisation and an increased risk of long-term mortality [4]. Additional independent correlates of mortality included remote and recent myocardial infarction, cerebral and peripheral vascular disease, chronic pulmonary disease, diabetes mellitus, dialysis and chronic renal disease [4]. The aforementioned risk factors were also represented in a significantly high percentage within our palliative study group.

The main objective of revascularisation in this population was to decrease the occurrence of angina pectoris and provide the survival benefit linked to LITA-LAD, while leaving other myocardial areas untreated [8,9].

In our previous publication, the actuarial survival rate of the 30-day survivors was 97%, 82% and 73% in an octogenarian population that underwent minimally invasive coronary surgery [5]. The current study reports survival rates of 83%, 67% and 61% at one, three and five years, respectively. The “palliative” population was marginally younger (median age 79.7 versus 83.2 years), but they had higher rates of chronic pulmonary disease, cerebral and peripheral vascular disease, medically treated type II diabetes, remote and recent myocardial infarction, dialysis and chronic renal disease. These factors account for the differences in mid-term survival. Furthermore, the octogenarian group had a larger percentage of two-vessel disease and a lower percentage of three-vessel disease. They also achieved a 69.8% completeness of revascularisation rate, which included 19.2% planned hybrid surgeries [5].

Rastan et al. examined a population of 8 806 consecutive MV CAD patients treated with CABG and concluded that reasonable incomplete revascularisation of the circumflex artery or right coronary artery, using LITA-to-LAD grafts, does not impact early and long-term survival [13]. The cumulative survival rates at one year and five years were comparable [13]. Despite this encouraging discovery, the authors came to the conclusion that surgeons should modify their surgical approach according to the risks of each patient rather than being misled into performing incomplete revascularisation in patients with graftable vessels [13].

Between 1997 and 2005, Jacob et al. performed MIDCAB on 80 high-risk patients with MV CAD [14]. For the population considered, the logistic EuroSCORE predicted a mortality rate of 10.2% [14]. With an 87-percent follow-up, they reported a 4-year survival rate of 85.6% and a MACCE-free survival rate of 81.5% [14]. The authors came to the conclusion that in certain patients with MV CAD, incomplete revascularisation using MIDCAB is a safe and successful technique [14].

Our work group opted for this customised strategy in high-risk and frail patients to lessen the trauma associated with thoracotomy, reduce narcotic consumption, facilitate early extubation and enable prompt postoperative mobilisation and recovery in patients who might otherwise face unfavourable outcomes with standard grafting methods. It is possible that the addition of routine frailty screening may offer patients with a more realistic outlook on their recovery, which will, in turn, improve the process of informed consent [11].

After analysing this research population, we suggest that a hybrid concept that includes a minimally invasive off-pump surgical approach could be an acceptable therapeutic choice in certain high-risk individuals.

### Limitations

One drawback of our study is that it was conducted on a small cohort of patients treated at a single centre over an extended duration. Another disadvantage is the inability to perform multivariate analysis owing to the low frequency of meaningful occurrences. Unfortunately, we did not use any frailty scales, therefore we do not have any records of how frail the patients were.

## 5. Conclusions

MIDCAB and MICS CABG can be successfully conducted as less invasive palliative surgery in high-risk multimorbid patients with MV CAD. The early and mid-term results were better than predicted. A higher rate of hybrid procedures could improve long-term outcome in selected cases.

## Figures and Tables

**Table 1 clinpract-15-00147-t001:** Baseline patient characteristics.

Variable	No (57)
Age (y)	79.7 ± 7.4
Male gender	46 (80.7)
BMI	24.9 ± 3.3
EuroSCORE II risk categories	
Very low risk < 1%	0
Low risk 1–2.99%	17 (29.8)
Moderate risk 3–4.99%	13 (22.8)
High risk > 5%	27 (47.3)
COPD	11 (19.3)
Current tobacco use	9 (15.8)
AF	26 (45.6)
PVD	16 (28.1)
Medically treated type II diabetes	18 (31.6)
CKD	25 (43.9)
Renal replacement therapy	4 (7)
Carotid stenosis (>60%)	12 (21.1)
Previous CVA	7 (12.3)
Previous MI	
<48 h	15 (26.3)
48 h–21 d	2 (3.5)
21 d–91 d	6 (10.5)
>91 d	2 (3.5)
Previous PCI	17 (29.8)
Coronary angiography data	
1 VD	3 (5.3)
2 VD	14 (24.5)
3 VD	40 (70.2)
Priority	
elective	47 (82.5)
urgent	10 (17.5)

Table 1: Values are presented as no (%) or mean ± SD (standard deviation); AF, atrial fibrillation; BMI, body mass index; CKD, chronic kidney disease; CVA, cerebrovascular accident; COPD, chronic pulmonary disease; EuroSCORE, European system for cardiac operative risk evaluation; GFR, glomerular filtration rate; LVEF, left ventricular ejection fraction; MI, myocardial infarction; PCI, percutaneous coronary intervention; PVD, peripheral vascular disease; VD, vessel disease.

**Table 2 clinpract-15-00147-t002:** Operative Data.

Variable	No (57)
Type of procedure	
MIDCAB	48 (84.2)
MICS CABG	9 (15.8)
Use of LITA	57 (100)
Use of RA	3 (5.3)
Use of GSV	1 (1.8)
Conversion to sternotomy and/or CPB	0
Number of grafts to the anterior wall	
1	51 (89.5)
2	6 (10.5)
Number of grafts to the lateral wall	
1	4 (7)
Complete revascularization	15 (26.3)
Planned as hybrid procedures	8 (14)

Table 2: Values are presented as no (%); CPB, cardiopulmonary bypass; GSV, great saphenous vein; LITA, left internal thoracic artery; MIDCAB, minimally invasive direct coronary artery bypass; MICS CABG, minimally invasive multivessel off-pump coronary artery bypass grafting; RA, radial artery.

**Table 3 clinpract-15-00147-t003:** Perioperative results.

Variable	No (57)
New renal failure requiring dialysis	6 (10.5)
Postoperative CVA	2 (3.5)
SSI	1 (1.8)
Postoperative CPR	3 (5.3)
Postoperative MI	1 (1.8)
Postoperative PCI as a revision	0
Postoperative PCI planned as part of hybrid procedure (30 d)	1 (1.8)
Reoperation for bleeding	0
Reoperation with bypass revision	0
Length of ICU stay (d)	2.8 ± 4.5
Length of hospital stay (d)	12.0 ± 5.7
30 d mortality	4 (7)

Values are presented as no (%) or mean ± SD (standard deviation); CPR, cardiopulmonary resuscitation; CVA, cerebrovascular accident; ICU, intensive care unit; MI, myocardial infarction; PCI, percutaneous coronary intervention; SSI, surgical site infection.

**Table 4 clinpract-15-00147-t004:** Late follow up results depending on rate of revascularization.

Variable	Total (57)	Incompletely Revascularise (42)	Completely Revascularised (15)	*p* Value
New renal failure requiring dialysis	6	2	4	0.539
Postoperative CVA	2	2	0	0.539
Postoperative MI	1	1	0	0.737
Late MI	6	4	2	0.504
Late PCI	8	5	3	0.419
Late CVA	1	1	0	0.712
Late MACCE	33	24	9	0.490
30 d mortality	4	2	2	0.281
Late mortality	24	18	6	0.399

Values are presented as no; CVA, cerebrovascular accident; MACCE, major adverse cardiac and cerebrovascular events; MI, myocardial infarction; PCI, percutaneous coronary intervention.

## Data Availability

The raw data supporting the conclusions of this article will be made available by the authors on request.

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
