# Peer review of "Minimally Invasive Off-Pump Coronary Artery Bypass as Palliative Revascularization in High-Risk Patients"

_clinpract, 2025, doi:10.3390/clinpract15080147_

Round 1
Reviewer 1 Report
Comments and Suggestions for Authors
I have been invited to review this manuscript, which analyzes 57 patients who underwent MIDCAB or MISC-CABG. The study was conducted on a very small sample over a period of 10 years, which makes it unsafe to draw any meaningful conclusions. I suggest that the authors compare the current cohort to patients with a similar profile who underwent full CABG, or consider including their cohort in a large international registry. At present, the study does not provide any added value.
Comments on the Quality of English LanguageEnglish needs to be improved.
Author Response
Comments 1: I have been invited to review this manuscript, which analyzes 57 patients who underwent MIDCAB or MISC-CABG. The study was conducted on a very small sample over a period of 10 years, which makes it unsafe to draw any meaningful conclusions. I suggest that the authors compare the current cohort to patients with a similar profile who underwent full CABG, or consider including their cohort in a large international registry. At present, the study does not provide any added value.
Response 1: We appreciate your dedication to reading and analysing our manuscript, as well as your expertise in offering such professional and comprehensive feedback. Your concerns are entirely valid, and we fully acknowledge them. The reason we decided to conduct this analysis in the first place was that we couldn't find a cohort of patients with a similar risk profile who underwent full CABG in the literature, but we did find a lot of literature, or case series from clinics, that shared our concern, namely how to better help these patients. The reports exhibited a range of approaches, from optimised medical therapy to restricted percutaneous coronary intervention, culminating in incomplete minimally invasive surgery as observed in our study group.
We fully endorse the idea of incorporating our data into a sizable international registry, and we believe that a meta-analysis of pooled data from comparable studies could potentially influence future "pallliative" case management. Additionally, it would undoubtedly have a statistically higher value while incorporating a greater number of patients from various centres.
Response 2: The English has been refined by a native speaker. Thank you for emphasizing this point and thereby enhancing the clarity of our paper for those interested in our findings.

Reviewer 2 Report
Comments and Suggestions for Authors
This is a study of 57 consecutive patients undergoing minimally invasive direct coronary artery bypass (MIDCAB) or minimally invasive multivessel coronary artery bypass grafting (MICS-CABG) as palliative surgery. The reason to be labeled as palliative is because the patients were very old. Their mean patient age was 79.7 ± 7.4 years and they had a lot of co-morbidities (AF and CKD). They were high risk for standard surgery so 48 underwent MIDCAB and 9 MICS-CABG. only 15 patients (26.3%) were completely revascularized. All patients had bypass with the left internal mammary artery (LIMA). Even the patients did not have complete revascularization, their survival was robust. Their one-, three- 32 and five-year survival rates were 83%, 67% and 61%, respectively.
The methodology was great, even it is a retroactive study.
The readers can learn a lot from this study.
Additional comments:
Please consider providing some more specific comments addressing the
following points: 3661137
1• What is the main question addressed by the research? Important question:
minimally invasive direct coronary artery bypass for high risk and elderly frail patients
2• Do you consider the topic original or relevant to the field? Does it
address a specific gap in the field? It is an important topic because there are not much data on elderly frail patients who had significant coronary disease.
- What does it add to the subject area compared with other published
material? It adds new data on how to treat elderly frail patients who had significant coronary disease.
- What specific improvements should the authors consider regarding the
methodology? None, because not every medical center has large number of elderly frail patients who had significant coronary disease.
- Are the conclusions consistent with the evidence and arguments presented
and do they address the main question posed? Please also explain why this
is/is not the case. YES
- Are the references appropriate? YES
- Any additional comments on the tables and figures. None
Author Response
Comments 1: This is a study of 57 consecutive patients undergoing minimally invasive direct coronary artery bypass (MIDCAB) or minimally invasive multivessel coronary artery bypass grafting (MICS-CABG) as palliative surgery. The reason to be labeled as palliative is because the patients were very old. Their mean patient age was 79.7 ± 7.4 years and they had a lot of co-morbidities (AF and CKD). They were high risk for standard surgery so 48 underwent MIDCAB and 9 MICS-CABG. only 15 patients (26.3%) were completely revascularized. All patients had bypass with the left internal mammary artery (LIMA). Even the patients did not have complete revascularization, their survival was robust. Their one-, three- 32 and five-year survival rates were 83%, 67% and 61%, respectively.
The methodology was great, even it is a retroactive study.
The readers can learn a lot from this study.
Additional comments:
Please consider providing some more specific comments addressing the
following points: 3661137
1• What is the main question addressed by the research? Important question:
minimally invasive direct coronary artery bypass for high risk and elderly frail patients
2• Do you consider the topic original or relevant to the field? Does it
address a specific gap in the field? It is an important topic because there are not much data on elderly frail patients who had significant coronary disease.
- What does it add to the subject area compared with other published
material? It adds new data on how to treat elderly frail patients who had significant coronary disease.
- What specific improvements should the authors consider regarding the
methodology? None, because not every medical center has large number of elderly frail patients who had significant coronary disease.
- Are the conclusions consistent with the evidence and arguments presented
and do they address the main question posed? Please also explain why this
is/is not the case. YES
- Are the references appropriate? YES
- Any additional comments on the tables and figures. None
Response 1: We appreciate your consideration in reviewing our paper. Your commitment to engaging with our manuscript and your proficiency in providing such thorough and insightful feedback are greatly valued.

Reviewer 3 Report
Comments and Suggestions for Authors
I woukd thank you the Editor for giving me the opportunity to revise the manuscript entitled "Minimally invasive off-pump coronary artery bypass as palliative revascularization in high-risk patients".
I red with interest this important topic because there is the need of some unanswered questions regarding myocardial revascularization by means of a less invasive approach such as off-pump and minimally invasive treatment.
Although in the discussion the Authors have tried to justify the absence of a frailty score, I think it is important to be able to objectively quantify the definition of high risk, not only with the Euroscore evaluation.
Furthermore, the percentage of patients undergoing complete revascularization was significantly low (26.3%) and no distinction was made with those who received incomplete revascularization in terms of mortality and short- and long-term complications.
In the tables presented it would be useful to indicate the total number of patients. In Table 1 the presence of carotid stenosis is indicated but not the degree (in terms of percentage).
Comments on the Quality of English LanguageI think that many of the concepts can be improved from a linguistic point of view.
Author Response
Comments 1: I woukd thank you the Editor for giving me the opportunity to revise the manuscript entitled "Minimally invasive off-pump coronary artery bypass as palliative revascularization in high-risk patients".
I red with interest this important topic because there is the need of some unanswered questions regarding myocardial revascularization by means of a less invasive approach such as off-pump and minimally invasive treatment.
Although in the discussion the Authors have tried to justify the absence of a frailty score, I think it is important to be able to objectively quantify the definition of high risk, not only with the Euroscore evaluation.
Response 1: Thank you for dedicating your time to read and analyse our text and for providing important feedback. You are quite correct, and we acknowledged this in the text as well as in the sections on limitations because it makes it difficult to extrapolate from or compare our findings to other research of a similar nature. For this reason, we made an effort to compare tangible statistics in the discussion section, such as mortality, the percentages of various comorbidities in our cohort compared to others, and even EuroSCORE II comparisons. Consequently, following the analysis and interpretation of this data, we have successfully integrated a frailty scale into our daily practice to standardise these cases.
Comment 2: Furthermore, the percentage of patients undergoing complete revascularization was significantly low (26.3%) and no distinction was made with those who received incomplete revascularization in terms of mortality and short- and long-term complications.
Response 2: We appreciate this insightful remark. A statistical analysis was conducted; however, due to the limited size of the patient cohort and the lack of significant differences between those with complete and incomplete revascularisation, we chose not to include it in the original publication. In accordance with your suggestions and request, we have incorporated an additional paragraph and a table into the 'Results' part of the paper.
Comment 3: In the tables presented it would be useful to indicate the total number of patients. In Table 1 the presence of carotid stenosis is indicated but not the degree (in terms of percentage).
Response 3: Thank you again for this observation. We have added the total number of patients to the tables and also indicated the degree of carotid stenosis.
Response 4: The English has been refined by a native speaker. Thank you for emphasizing this point and thereby enhancing the clarity of our paper for those interested in our findings!

Reviewer 4 Report
Comments and Suggestions for Authors
Article is well written with interesting subject and interesting approach of resolving revascularization problems in frail patients, generally well written.
There are some issues to be resolved and because they are spread across entire article I will address issue by issue not section by section. Other parts are very well described with lots of details and 4 point MACE is used what is commendable.
- Number of cases/patients:
In materials and methods authors stated “we identified 57 cases treated between September 2008 and December 2018.
In results reported: There were four postoperative fatalities... A total of 52 patients were monitored, yielding a completion rate of 91.2%....
52 monitored + 4 postoperative fatalities = 56 patients; 1 missing – please clarify?
- Study goal is not clearly defined:
Introduction: This study aims to explore the early and late clinical outcomes following MIDCAB and MICS-CABG in a consecutive cohort of high-risk patients at a single center.
Materials and methods: The primary objective of this research was to examine the all-cause death rate in the early and midterm periods. The secondary aim included evaluating the incidence of major adverse cerebral and cardiovascular events (MACCE) during the follow-up interval.
Results: There were four postoperative fatalities... The follow-up time on average was 4.2 ± 3.7 years. A total of 52 patients were monitored, yielding a completion rate of 91.2%. Survival rates at one-, three-, and five-years were 83%, 67%, and 61%, respectively. During the follow-up period, 33 major adverse cardiac and cerebrovascular events were recorded. Six myocardial infarctions, eight repeat revascularizations via PTCA, one stroke, and 24 fatalities constituted the causes of MACCE. Only two fatalities were confirmed to have a cardiac cause.
Mortality (all-cause of death) rate is not same thing as survival rates, for mortality study this one is seriously underpowered and I suggest authors to redefine primary goal in Materials and methods section. In results section do to small number of cases I suggest to use exact number of patients, not percentages.
- Decision making process of Heart team.
Authors should more precisely describe decision making process in patients treatment by Heart team (algorithm would be acceptable), which criteria were used to choose between therapeutic options (medicaments therapy only, PCI, hybrid procedure, surgery – described methods). The only criteria authors mentioned was “solely on coronary angiogram assessment” what we know is not enough based on many studies. Were vascular imaging methods and/or hemodynamic methods used to asses vessel/stenosis, was microcirculation assessed ? If those tools were not available or used it should be stated in study limitations.
- Statistics: For Kaplan-Meier test p value is missing.
- Term "palliative" is used in context of revascularization (This revascularisation was classified as palliative) and population (The "palliative" population...). Please clearly define meaning – patient general condition and/or coronary artery condition?
- Table 1. has several issues:
EuroSCORE II = 7.3 ± 6.8 – standard deviation is very high (score range is 0.5-14.1), I suggest that patients should be stratified in risk groups according to score (very low risk, low risk, moderate risk, high risk)
Tobacco use: are those current smokers or former + current smokers, please clarify ?
Coronary angiography data: 1, 2 or 3 vessel disease is presented in table; I suggest that vessels should be specified what would give additional value to article (not obligatory as authors already described that most procedures were to anterior wall – LAD or LMCA is about in most of cases, understandable to interventional cardiologist but may not to other readers).
EFLV (%) classification > 50 • 30-50 • < 30; 28 (49.1) • 22 (38.6) • 7 (12.3). This classification is used to calculate EUROSCORE II, but is not for classification of EFLV according to ECS guidelines for heart failure with preserved, mild reduced or reduced ejection fraction. Please correct this according to ESC guidelines if you want to present it correctly (in example patient in line 2 in table can have preserved, mild reduced or reduced ejection fraction) Or since EFLV is already calculated in EUROSCORE II could be left out.
Author Response
Comment 1: Article is well written with interesting subject and interesting approach of resolving revascularization problems in frail patients, generally well written.
There are some issues to be resolved and because they are spread across entire article I will address issue by issue not section by section. Other parts are very well described with lots of details and 4 point MACE is used what is commendable.
- Number of cases/patients:
In materials and methods authors stated “we identified 57 cases treated between September 2008 and December 2018.
In results reported: There were four postoperative fatalities... A total of 52 patients were monitored, yielding a completion rate of 91.2%....
52 monitored + 4 postoperative fatalities = 56 patients; 1 missing – please clarify?
Response 1: Thank you for this insightful observation. The study comprised 57 cases, of which 4 fatalities occurred within the first 30 days. Monitoring continued for 48 cases, while 5 were unreachable and thus considered lost to follow-up. This explanation has been rephrased and incorporated throughout the text.
Comment 2:
- Study goal is not clearly defined:
Introduction: This study aims to explore the early and late clinical outcomes following MIDCAB and MICS-CABG in a consecutive cohort of high-risk patients at a single center.
Materials and methods: The primary objective of this research was to examine the all-cause death rate in the early and midterm periods. The secondary aim included evaluating the incidence of major adverse cerebral and cardiovascular events (MACCE) during the follow-up interval.
Results: There were four postoperative fatalities... The follow-up time on average was 4.2 ± 3.7 years. A total of 52 patients were monitored, yielding a completion rate of 91.2%. Survival rates at one-, three-, and five-years were 83%, 67%, and 61%, respectively. During the follow-up period, 33 major adverse cardiac and cerebrovascular events were recorded. Six myocardial infarctions, eight repeat revascularizations via PTCA, one stroke, and 24 fatalities constituted the causes of MACCE. Only two fatalities were confirmed to have a cardiac cause.
Mortality (all-cause of death) rate is not same thing as survival rates, for mortality study this one is seriously underpowered and I suggest authors to redefine primary goal in Materials and methods section. In results section do to small number of cases I suggest to use exact number of patients, not percentages.
Response 2: Thank you for these very good observations. We have changed according to your suggestion the definition of the aim of the study and also the reports on survival.
Comment 3:
- Decision making process of Heart team.
Authors should more precisely describe decision making process in patients treatment by Heart team (algorithm would be acceptable), which criteria were used to choose between therapeutic options (medicaments therapy only, PCI, hybrid procedure, surgery – described methods). The only criteria authors mentioned was “solely on coronary angiogram assessment” what we know is not enough based on many studies. Were vascular imaging methods and/or hemodynamic methods used to asses vessel/stenosis, was microcirculation assessed ? If those tools were not available or used it should be stated in study limitations.
Response 3: Thank you for your keen observation. We have rewritten the Methods chapter as to make it more understandable to the reader.
Comment 4:
- Statistics: For Kaplan-Meier test p value is missing.
Response 4: Thank you for this comment. The Kaplan Meier test did not generate a p value as we don’t have a control group.
Comment 5:
- Term "palliative" is used in context of revascularization (This revascularisation was classified as palliative) and population (The "palliative" population...). Please clearly define meaning – patient general condition and/or coronary artery condition?
Response 5: Thank you for this important comment. We have defined as explained this term in the revised version of the paper.
Comment 6:
- Table 1. has several issues:
EuroSCORE II = 7.3 ± 6.8 – standard deviation is very high (score range is 0.5-14.1), I suggest that patients should be stratified in risk groups according to score (very low risk, low risk, moderate risk, high risk)
Tobacco use: are those current smokers or former + current smokers, please clarify ?
Coronary angiography data: 1, 2 or 3 vessel disease is presented in table; I suggest that vessels should be specified what would give additional value to article (not obligatory as authors already described that most procedures were to anterior wall – LAD or LMCA is about in most of cases, understandable to interventional cardiologist but may not to other readers).
EFLV (%) classification > 50 • 30-50 • < 30; 28 (49.1) • 22 (38.6) • 7 (12.3). This classification is used to calculate EUROSCORE II, but is not for classification of EFLV according to ECS guidelines for heart failure with preserved, mild reduced or reduced ejection fraction. Please correct this according to ESC guidelines if you want to present it correctly (in example patient in line 2 in table can have preserved, mild reduced or reduced ejection fraction) Or since EFLV is already calculated in EUROSCORE II could be left out.
Response 6: Thank you for these valuable comments. We have changed and rewritten these variables in table 1 accordingly.
Round 2
Reviewer 1 Report
Comments and Suggestions for Authors
I maintain my position from the previous review, given the small cohort size, the long follow-up period, and the fact that nearly 10% of patients were lost to follow-up, it is not possible to draw meaningful conclusions
Reviewer 3 Report
Comments and Suggestions for Authors
Dear Authors,
thank you for your reply and for the implementations reported in the manuscript. Although the very limited number of patients examined in the study (as clearly explained in the "Limitation" paragraph) does not constitute a strong contribution to the surgical strategy it will be useful in arousing the interest and future study by those who daily deal with this complex group of patients.